# Assay Development and Validation for Innovative Antiviral Development Targeting the N-Terminal Autoprocessing of SARS-CoV-2 Main Protease Precursors

**DOI:** 10.3390/v16081218

**Published:** 2024-07-29

**Authors:** Liangqun Huang, Megan Gish, James Boehlke, Ryan H. Jeep, Chaoping Chen

**Affiliations:** Department of Biochemistry and Molecular Biology, Colorado State University, Fort Collins, CO 80523, USA; liangqun.huang@colostate.edu (L.H.); meganegish@gmail.com (M.G.); james1boehlke@gmail.com (J.B.); ryan.jeep@colostate.edu (R.H.J.)

**Keywords:** autoproteolysis, 3C-like protease, catalysis flexibility, glycosylation, main protease, Nsp5, precursor autoprocessing, protease precursor, SARS-CoV-2

## Abstract

The SARS-CoV-2 main protease (M^pro^) is initially synthesized as part of polyprotein precursors that undergo autoproteolysis to release the free mature M^pro^. To investigate the autoprocessing mechanism in transfected mammalian cells, we examined several fusion precursors, with the mature SARS-CoV-2 M^pro^ along with the flanking amino acids (to keep the native substrate sequences) sandwiched between different tags. Our analyses revealed differential proteolysis kinetics at the N- and C-terminal cleavage sites. Particularly, N-terminal processing is differentially influenced by various upstream fusion tags (GST, sGST, CD63, and Nsp4) and amino acid variations at the N-terminal P1 position, suggesting that precursor catalysis is flexible and subject to complex regulation. Mutating Q to E at the N-terminal P1 position altered both precursor catalysis and the properties of the released M^pro^. Interestingly, the wild-type precursors exhibited different enzymatic activities compared to those of the released M^pro^, displaying much lower susceptibility to known inhibitors targeting the mature form. These findings suggest the precursors as alternative targets for antiviral development. Accordingly, we developed and validated a high-throughput screening (HTS)-compatible platform for functional screening of compounds targeting either the N-terminal processing of the SARS-CoV-2 M^pro^ precursor autoprocessing or the released mature M^pro^ through different mechanisms of action.

## 1. Introduction

Coronaviruses are positive-sense single-stranded RNA (+ssRNA) viruses belonging to the *Coronaviridae* family, capable of infecting a wide array of mammalian and avian species [1,2,3,4,5]. Before the 21st century, four coronaviruses (HCoV-229E, -NL63, -OC43, and -HKU1) were known to cause mild common colds in humans, with more severe disease occurring in infants, the elderly, and immunocompromised individuals [1]. Since then, three additional coronaviruses have crossed the species barrier to cause lethal pneumonia in humans: severe acute respiratory syndrome coronavirus (SARS-CoV) [2], Middle East respiratory syndrome coronavirus (MERS-CoV) [3], and severe acute respiratory syndrome coronavirus 2 (SARS-CoV-2) [4,5]. These viruses, along with other +ssRNA viruses, encode a main protease (M^pro^), also known as 3C or 3C-like proteases (3CL^pro^) or Nsp5, which is crucial for viral replication [6,7]. In infected cells, M^pro^ is initially synthesized as part of the polyprotein precursors pp1a or pp1ab (Figure 1A). The liberation of the free mature M^pro^ is mediated by the still-embedded M^pro^ through precursor autoprocessing, although the underlying mechanism remains largely elusive [8,9]. The mature M^pro^ cleaves polyprotein precursors at up to 11 diverse substrate sites, involving a conserved glutamine (Gln) at the P1 position and a small amino acid (Ser, Ala, or Gly) at the P1′ position with varying efficiencies [10,11,12,13].

Significant research efforts have focused on the structural characterization of mature M^pro^ [14,15,16,17,18] and structure-guided inhibitor development [17,19,20,21,22,23,24,25]. These studies reveal that SARS-CoV-2 M^pro^ is a cysteine protease with a chymotrypsin-like fold. Dimerization of recombinant mature M^pro^, expressed in and purified from *E. coli*, is required for M^pro^ catalysis, but each protomer’s active site is not directly involved in dimerization [17,18,26,27,28]. In general, it is believed that dimerization contributes to stabilization of the active sites that recognize various substrates with diverse sequences [27,29,30]. Extensive structural investigations have led to the development of several patented inhibitors [17,21], including GC376 [31,32], which laid the groundwork for the development of nirmatrelvir, the first orally active M^pro^ inhibitor developed by Pfizer and marketed as Paxlovid (nirmatrelvir/ritonavir combination) [33,34,35,36]. Both GC376 and nirmatrelvir contain a rigid pyrrolidone mimicking the glutamine side chain at the P1 position for competitive binding. Recent molecular dynamics simulations have indicated conformational flexibility and plasticity of the catalytic site involved in substrate binding, which poses challenges for structure-based small-molecule inhibitor designs [37]. The emergence of nirmatrelvir-resistant mutations also necessitates innovative antiviral development [38,39,40].

In contrast to the extensive investigations on mature M^pro^ structure and function, characterization of M^pro^ precursor autoprocessing has been limited and has exclusively used recombinant proteins expressed in and purified from *E. coli* [8,10,11,12,26,29,41,42,43]. Therefore, it remains unclear how M^pro^ precursors autoprocess inside mammalian cells. In this study, we examined the autoprocessing of fusion precursors carrying the mature M^pro^, along with flanking amino acids to preserve the cleavage sites, sandwiched between fusion tags. This assay platform allowed us to investigate the effects of upstream fusion tags and mutations at the N- or C-terminal cleavage sites (particularly the conserved Gln at the P1 position) on precursor autoprocessing in transfected HEK293T cells. Our results revealed several intriguing aspects of precursor catalysis and its complex regulation, demonstrating that: (1) precursor catalysis appears sensitive to regulation: different upstream fusion proteins or variations of P1 residue can influence autoprocessing efficiency; (2) the M^pro^ released from different precursors displays distinct enzymatic properties: self-degradation prone from the wt precursor, whereas self-degradation resistant from the precursor with mutation Q to E at the N-terminal P1 position; (3) precursors are less sensitive than the released M^pro^ to suppression by several currently available M^pro^ inhibitors, suggesting that they are enzymatically different from each other. Consequently, precursors could serve as alternative targets for antiviral development. We also established a high-throughput screening platform to enable functional screening for compounds targeting both the precursor and released M^pro^ through different mechanisms of action.

## 2. Materials and Methods

Plasmid construction—All mammalian expression vectors used in this study were engineered through standard PCR-based mutagenesis, molecular cloning techniques, and confirmed by DNA sequencing. Detailed information for each plasmid is described in the corresponding result section, as precursor autoproteolysis is sensitive to modulation by various fusion contexts. The sGST-fused HIV-1 precursors for mammalian expression driven by the human EF-1α promoter were described previously [44,45,46]. Genomic RNA from SARS-CoV-2 (USA-WA1/2021, BEI NR-52285 Manassas, VA, USA) or MERS-CoV (EMC/2012, BEI NR-45843) was used to generate cDNA templates containing the corresponding M^pro^ coding sequences for PCR amplification. The resulting PCR fragments carrying SARS-CoV-2 or MERS-CoV M^pro^ sequences, along with N- or C-terminal amino acids and engineered restriction cut sites, were digested, purified, and ligated into GST or sGST plasmid in place of the HIV-1 precursor coding sequence. CD63- and Nsp4-fused constructs were generated by replacing the GST coding sequence in the corresponding SARS-CoV-2 M^pro^ expressing vector. A codon-optimized bacterial SARS-CoV-2 Nsp4 expression vector (BEI, NR-53497) was used as a PCR template. These plasmids are available with a standard material transfer agreement.

Inhibitors and APExBio Protease Inhibitor Library—Inhibitors GC376 and nirmatrelvir (PF-07321332) were purchased from AOBIOS (cat# AOB36447 and AOB14800, respectively, Gloucester, MA, USA). Boceprevir and calpain inhibitor II were purchased from VWR (cat# 76408-330 and 10191-132, respectively, Radnor, PA, USA). Compounds Z-FA-FMK and E-64 were purchased from APExBio (cat# A8170 and A2576, respectively, Houston, TX, USA). HIV-1 protease inhibitors atazanavir (cat# HRP-10003) and lopinavir (cat# HRP-9481) were obtained through the NIH AIDS Reagent Program, Division of AIDS, NIAID, NIH (Bethesda, MD, USA). The APExBio Protease Inhibitor Library (cat# L1035, APExBio) contains a collection of 130 protease inhibitors dissolved in DMSO at 10 mM (100 µL/well in one and a half 96-well plates). Detailed information is available online “https://www.apexbt.com/downloader/panellist/L1035-DiscoveryProbe-Protease-Inhibitor-Library.xlsx (accessed on 22 July 2024)”.

Cell Culture and Transfection—HEK 293T cells purchased from ATCC (cat# CRL-11268, Manassas, VA, USA) were cultured in DMEM medium containing 10% fetal bovine serum, 100 units/mL of penicillin G sodium salt, and 100 µg/mL streptomycin sulfate. The calcium phosphate transfection procedure was described previously [44,45,46,47,48]. In brief, cells were seeded in a 12-well plate one day prior to transfection, aiming for 30–40% confluence at the time of transfection. Each well first received chloroquine to a final concentration of 25 µM. A DNA–calcium mixture was made by mixing 0.5 µg plasmid DNA in 65.7 µL H_2_O with 9.3 µL of 2 M CaCl_2_. Then, 75 µL of 2 × HBS (50 mM Hepes, pH 7.04–7.05, 10 mM KCl, 12 mM dextrose, 280 mM NaCl, and 1.5 mM Na_2_HPO_4_) was added dropwise to the DNA–calcium mixture with gentle vortexing. Each well received the final mixture dropwise. At 7–11 h post-transfection, fresh chloroquine-free medium was added to remove chloroquine. For drug treatment, inhibitors at the indicated concentrations were added at this time. At ~30 h post-transfection, the medium was removed, and cells were gently rinsed with 1× PBS and lysed in situ with 80 µL lysis buffer A (Tris-HCl, pH 8.0, 150 mM NaCl, 1% sodium deoxycholate, 1% Triton X-100, and 1× protease inhibitor cocktail). Under these conditions, the cellular chromosomes formed viscous and filamentous aggregates that were removed with a pipette tip. The remaining post-nuclear lysate was collected for SDS-PAGE or stored at −20 °C until use.

SDS-PAGE and Western Blotting—Cell lysates were denatured at 95 °C for 5 min, followed by a two-minute chill on ice, one-minute centrifugation at 3000× *g*, and a brief vortex prior to SDS-PAGE analysis. Proteins resolved through a homemade 8–18% gradient gel were then transferred onto a PVDF membrane (Fisher, cat# IPVH00010, Waltham, MA, USA) and probed with antibodies. Primary antibodies used in this study include rabbit anti-GST (Sigma, cat# G7781) at 1:1000, mouse anti-flag (Sigma-Aldrich, cat# F1804, Burlington, MA, USA) at 1:1000, mouse monoclonal anti-HA (Sigma, cat# H9658) at 1:10,000, mouse anti-GAPDH (Millipore, cat# MAB374) at 1:25,000, mouse anti-GFP (Rockland Immunochemicals Inc., cat# 600-301-215, Pottstown, PA, USA) at 1:1000, and mouse anti-CD63 (LAMP3) (Santa Cruz Biotechnology, cat# SC-5275, Santa Cruz CA, USA) at 1:250. Secondary antibodies included IR700 fluorescence-labeled goat anti-rabbit (Rockland, cat# 611-130-122) and IR800 goat anti-mouse (Rockland, cat# 610-132-121). The blots were visualized with an Odyssey infrared dual-laser scanning unit (LI-COR Biotechnology, Lincoln, NE, USA). To reduce background noise, some primary antibody solutions were pre-absorbed against cell lysates made from untransfected cells that were resolved by SDS-PAGE and transferred onto a PVDF membrane. Image Studio version 5.2 software was used to determine band intensity by the total pixel value, subtracting the background noise in the same area. Band intensity representing the detection protein was used to calculate autoprocessing efficiency and mature M^pro^ detection.

Deglycosylation of cell lysates by PNGase—Transfected HEK 293T cells in a 6-well plate were lysed with 150 µL lysis buffer A (25 mM Tris-HCL, pH 8.0, 150 mM NaCl, 1% sodium deoxycholate, 1% Triton X-100, plus protease inhibitor cocktail) per well. The lysates were collected and spun at 1000× *g* for 2 min to remove debris and nuclei. The clarified supernatant was designated as post-nucleus lysates (also called inputs). PNGase F (New England Biolabs P0708S, Ipswich, MA, USA)-mediated deglycosylation was carried out per the manufacturer’s recommendation. In brief, the lysates were first denatured by heating at 100 °C for 10 min in glycoprotein denature buffer followed by a quick chill on ice. A total reaction volume of 20 µL was prepared by combining 10 µL denatured lysates, 2 µL GlycoBuffer 2 (10×), 2 µL 10% NP-40, 0.2–1 µL PNGase F, and H_2_O to make up the volume. The mixture was incubated at 37 °C for 1 h before mixing with 6 × SDS loading buffer for SDS-PAGE analysis.

Dual high-throughput screening—HEK293T grown in 10 cm plates were used for bulk transfection with the indicated fusion precursor construct plus a GFP-encoding plasmid accounting for ~3% of total DNA. At 7–11 h post-transfection, the transfected cells were lifted by trypsin-EDTA and seeded into a 384-well plate (Cat# 82051-278, Breiner BioOne, Monroe, NC, USA) at ~20–40% confluency. Inhibitors at the indicated concentrations were added to individual wells for treatment. All drugs were dissolved in DMSO at varying stock concentrations. Previous reports have indicated that DMSO has multiple inhibitory effects on assay performance when its final concentrations exceed 0.25%. Consequently, vehicle controls were included at the DMSO concentration matching the compound with the lowest dilution. For example, when testing the APExBio library at 20 µM, DMSO at 0.2% (500-fold dilution from the 10 mM stock) was included as the basal level control. After ~24 h of incubation, the culture medium was removed by gentle aspiration, and the remaining cells were lysed in situ by 20 µL assay solution containing 1× AlphaLISA Immunoassay buffer (PerkinElmer AL000F, Waltham, MA, USA), protease inhibitor cocktail, 15 µg/mL anti-flag acceptor beads (PerkinElmer AL112R), and 11.25 µg/mL glutathione donor beads (PerkinElmer 6765301). As a positive control, the purified protein GST-Flag-p6* made in 1× AlphaLISA Immunoassay buffer at 5-fold serial dilutions (8 points total) was spiked into wells only treated with DMSO. Following at least another 2-h incubation at 37 °C, each plate was read for fluorescence signal first, followed by AlphaLISA signal detection using an EnSpire Microplate Reader (PerkinElmer).

## 3. Results

### 3.1. Fusion M^pro^ Precursors Are Autoprocessing-Competent in Transfected Mammalian Cells

In the infected cell, M^pro^ is initially translated as part of the polyproteins 1a and 1ab (Figure 1A). Precursor autoproteolysis at two cleavage sites, Nsp4↓M^pro^ and M^pro^↓Nsp6, liberates free mature M^pro^, which is indispensable for productive viral replication. To study these autoproteolytic reactions in transfected mammalian cells, we engineered fusion precursor expression plasmids containing the mature M^pro^ along with the flanking residues (P8-P1 derived from Nsp4 and P1′-P8′ from Nsp6), thereby maintaining both the N- and C-terminal cleavage sites, sandwiched between the GST and Flag tags (Figure 1A). With both native cleavage sites present, the precursor could undergo different routes, producing different intermediates but ultimately yielding the same final products. Most of these proteins are detectable by Western blotting, except for the free mature M^pro^, due to the lack of an available antibody, and the P1′-P8′-Flag fragment, which is too small (~2 kD) for detection. These WB analyses via GST- or Flag-antibodies provided a snapshot of precursor autoprocessing in transfected cells. With the wild-type precursor, only GST-8 was detected in the cell lysates; no FL or intermediates were detectable (Figure 1B, lane 13), suggesting nearly complete proteolysis at both sites. Mutation C145G abolished autoproteolysis activity (Figure 1B, lanes 7 and 12), showing the unprocessed FL precursor as the only detectable band in both the GST and Flag channels, validating that the observed autoproteolysis was mediated by the fusion precursors and not by cellular proteases.

### 3.2. Differential Catalytic Kinetics at the N- and C-Terminal Autoprocessing Sites

Extensive sequence analysis has unveiled the notable conservation of Q at the P1 position among the substrates recognized and processed by the M^pro^ of SARS-CoV and other coronaviruses [10,11,12]. To investigate the role of these conserved Qs in precursor autoprocessing, we designed a series of variants with the conserved Q mutated to glutamate (E) individually or in combination. For the precursor Q_E8, detection of M^pro^-Flag indicated that the mutation of Q to E at the C-terminal P1 site effectively blocked C-terminal autoproteolysis (Figure 1B, lane 14). Meanwhile, N-terminal processing appeared unaffected, with nearly equal amounts of GST-8 detected compared to the wild-type Q_Q8 and minimal detection of the FL precursor. The precursor E_Q8 demonstrated the intricate role of the N-terminal P1 residue in the modulation of precursor autoprocessing. First, mutating Q to E did not abolish N-terminal processing, as evidenced by the detection of GST-8 (Figure 1B, lane 15), indicating that the conserved Q at this site may not be essential for cleavage. This contrasts with C-terminal autoprocessing, which relies heavily on the conserved Q at the P1 position for an optimal substrate. Second, mutating Q to E at the N-terminal P1 position reduced precursor catalysis, evidenced by increased detection of both the FL precursor (reactive to both GST and Flag antibodies) and GST-8-M^pro^ (reactive only to the GST antibody) (Figure 1B, lane 15). Detection of the FL precursor indicated incomplete C-terminal processing even with the wild-type substrate sequence at the C-terminus. GST-8-M^pro^ is an intermediate product of the FL precursor undergoing C-terminal processing first; its detection at steady-state also suggested less efficient subsequent N-terminal processing (indicated by the orange upward arrow in Figure 1A). Hence, the Q-to-E mutation at the N-terminal P1 position diminished catalysis of both the FL precursor and the GST-8-M^pro^ intermediate.

The E_E9 double mutation not only abolished C-terminal processing but further reduced N-terminal processing, as evidenced by increased detection of the FL precursor, along with the detection of GST-8 and M^pro^-Flag, two N-terminal processing products (Figure 1B, lane 16). Taken together, our data revealed distinct autoprocessing kinetics at the N- and C- terminal cleavage sites. Mutating Q to E at the C-terminal P1 abolished C-terminal processing while minimally affecting N-terminal processing and overall precursor catalysis. In contrast, the same mutation at the N-terminal P1 position did not prevent N-terminal processing but reduced autoproteolysis at both N- and C-terminal sites by the resulting precursor. Our findings suggest that precursor catalysis is intricately regulated beyond its active site, as the N-terminal P1 residue is not part of mature M^pro^.

We also engineered and analyzed a panel of X_X precursors where the Nsp6-derived P1′–P8′ residues were replaced with the VDYKDDDDK sequence (Figure 1A). This direct fusion of the Flag tag prevented C-terminal processing, even with the conserved Q at the C-terminal P1 position, as evidenced by the detection of M^pro^-Flag (Figure 1B, lanes 3 and 5, Flag blot). In this context, mutating Q to E at the C-terminal P1 position had no additional impact since the site was already uncleavable (Figure 1B, lanes 3–6, Flag blot). These data confirmed that C-terminal processing can be blocked either by altering P1 alone (Q to E) or by mutating P1′–P8′. Consequently, these X_X precursors provided a simple tool for specifically studying N-terminal autoprocessing. Within this context, mutating Q to E at the N-terminal P1 position did not block N-terminal processing but reduced precursor catalysis, similar to observations with the X_X8 precursors. Quantification of band intensity enabled us to compare autoprocessing efficiencies using the ratio of GST-8/(GST-8 + FL) through the GST channel or M^pro^-Flag/(M^pro^-Flag + FL) via the Flag channel, as proxies (Figure 1C). This quantitative analysis showed that the Q-to-E mutation at the N-terminal P1 position reduced autoprocessing efficiency from ~100% to ~70% via the GST channel in both X_X and X_X8 contexts. Similar results were obtained via the Flag channel except for the constructs with native C-terminal cleavage site (X_Q8), where detection of the released M^pro^ was unfeasible due to unavailability of the antibody. Collectively, the results with X_X precursors further confirmed a regulatory role of the N-terminal P1 residue on precursor catalysis, beyond merely being part of a cleavage site, and validated these X_X precursors for more focused characterization of N-terminal autoprocessing.

Additionally, we engineered and examined a panel of MERS-CoV M^pro^ constructs in the X_X context (Figure 1B, lanes 8–11) to determine whether the observed regulatory role of the N-terminal P1 residue with SARS-CoV-2 is common among coronaviruses. The wild-type MERS-CoV M^pro^ precursor was fully competent at N-terminal processing, as indicated by the detection of GST-8 and the absence of the full-length precursor (Figure 1B, lane 8). The released MERS-CoV M^pro^-Flag was below the detection limit, suggesting its rapid self-degradation after liberation. Interestingly, mutating Q to E at the N-terminal P1 position completely abolished MERS-CoV N-terminal processing, showing a detection pattern like that of the G145G control. These results illustrated different impacts of the Q-to-E mutation at the N-terminal P1 position, slightly reducing SARS-CoV-2 M^pro^ precursor catalysis but abolishing MERS-CoV M^pro^ precursor N-terminal autoprocessing. The observed differences between SARS-CoV-2 M^pro^ and the MERS-CoV M^pro^ precursor revealed additional complexity in precursor catalysis and its regulation. Subsequently, we focused on N-terminal autoprocessing of the SARS-CoV-2 M^pro^ precursor for characterization and antiviral development.

### 3.3. Modulation of N-Terminal Processing by N-Terminal P1 Residue Variations

Based on our results showing modulation of SARS-CoV-2 precursor catalysis by mutating Q to E at the N-terminal P1 position, we further examined the effects of other amino acids in the context of the GST-fused X_Q precursor (Figure 2A). N-terminal processing efficiencies, measured via the GST channel, exhibited a continuum of processing efficiencies (Figure 2B, upper panel), suggesting diverse substrate flexibility and complex regulation of precursor catalysis. Substitutions with histidine (H) or methionine (M) exhibited wild-type activity, indicating that both residues can support wild-type precursor activity and concurrently serve as proper substrates. The preference for H aligns with a report showing that recombinant M^pro^ effectively processes cellular proteins with H at the P1 position [49]. The preference for M is supported by the crystal structure of mature M^pro^ in a complex with calpain inhibitor II, which shows good accommodation of the M side chain in the S1 pocket [50]. Identification of these amino acids (Q, H, and M) as preferred P1 residues is valuable for the rational design of compounds to improve binding affinity if needed. Other variations, including E, also supported N-terminal processing but with reduced efficiencies compared to those of the top three preferred residues. In contrast, amino acids I, V, and P abolished N-terminal processing similarly to the C145G mutation. A common feature among these three amino acids is their unique C_β_ branching configurations, which may create steric hindrance when interfacing with the catalytic site, making them non-optimal substrates. Alterations to D, G, and W, along with the basic amino acids (K and R), also drastically decreased N-terminal processing. Although it remains to be determined whether these P1 variations occur in nature, it is interesting to note that the M^pro^ coding region along with its processing site is reported to have high mutation rates, which might contribute to drug resistance development [51,52].

Quantitative analysis via the Flag channel was generally consistent with the GST channel results, except for reduced M^pro^-Flag detection for many N-terminal P1 variations. While reduced M^pro^-Flag detection was expected for those that abolish N-terminal processing, the low detection levels of M^pro^-Flag, released from variations with moderate N-terminal autoprocessing efficiencies, suggest distinct catalysis kinetics of M^pro^-Flag products. One plausible explanation is differences in self-degradation propensity, with some rapidly degrading themselves, like MERS-CoV M^pro^-Flag (Figure 1B, lane 8), whereas others were less prone to self-degradation. In summary, while the conserved Q at the N-terminal P1 position is preferred, it is not essential; many amino acids are acceptable as part of cleavable substrates. This broad P1 residue specificity indicates the conformational flexibility of the precursor catalytic site, capable of accommodating diverse substrate peptides. This substrate flexibility explains why many peptidomimetic compounds with diverse functional groups at the P1 position can bind to the active site with various affinities. These findings provide insights into potential chemical scaffolds that can enhance binding affinities, aiding the rational design and development of novel M^pro^ inhibitors. Additionally, our findings highlight the intricate role of the N-terminal P1 residue in modulating the catalysis of both the precursor and the released M^pro^-flag, warranting further mechanistic investigation.

### 3.4. Modulation of N-Terminal Processing by Upstream Fusion Tags That Mediate Membrane Association

Given our findings that precursor catalysis can be influenced by mutations outside the mature M^pro^ coding region, we engineered a panel of sGST-fused precursors to assess whether N-terminal processing can also be affected by upstream fusion tags. The sGST fusion tag includes an MBP-derived signal peptide upstream of GST (Figure 3A), previously shown to mediate vesicle membrane targeting of fusion proteins in transfected HeLa cells [45], whereas GST-fused precursors are primarily cytosolic. Intriguingly enough, our data revealed that sGST-fused precursors were consistently less effective at N-terminal processing than their GST-fused counterparts. For example, all sGST-fused SARS-CoV-2 constructs with the conserved Q at the N-terminal P1 position (Q_M) demonstrated ~40% autoprocessing efficiencies via GST quantification, regardless of C-terminal P1 residue variations (Figure 3B,C). In contrast, GST-fused Q_X precursors were nearly 100% processed (Figure 1C). The sGST-fused MERS-CoV precursors with Q at the N-terminal P1 (Q_X) averaged only ~20% autoprocessing efficiencies (Figure 3C). These results indicated a negative modulation of M^pro^ N-terminal autoprocessing by the sGST fusion, suggesting that precursor catalysis is flexible and can be regulated in the context of polyprotein precursors. Mutating Q to E at the N-terminal P1 position further reduced N-terminal processing by ~30% (from ~40% to ~10%) in the sGST fusion context. Overall, our data confirm the modulation of precursor catalysis by both N-terminal P1 residue variation and upstream fusion tags.

Within polyproteins 1a and 1ab, M^pro^ (Nsp5) is preceded by Nsp4, a transmembrane protein with a tetraspanin configuration (having four transmembrane domains, with both N- and C- termini exposed to the cytoplasm). This positioning places M^pro^ in a subcellular environment near the ER membrane as the polyprotein is translated. Given our findings showing precursor autoprocessing subject to regulation by the fusion context, we further examined the effects of membrane proximity. We first used CD63, a well-characterized tetraspanin, as a fusion tag for the study. CD63 is known to undergo complex N-linked glycosylation, resulting in a smearing detection pattern (Figure 4B, lane 1 of the bottom panel) in mock-transfected cells. An N-terminal GFP fusion did not alter this smearing pattern (Figure 4B, lane 2). To minimize complications caused by these smearing bands for band detection and quantification, we treated the lysates with PNGase F, a broad-spectrum enzyme, that removes nearly all N-linked oligosaccharides by cleaving between the innermost GlcNAc and asparagine residues. After PNGase F deglycosylation, both GFP and CD63 antibodies detected a defined band with an apparent MW of ~40 kDa, which is ~20 kDa shorter than that of the calculated FL GFP-CD63 fusion protein (~60 kDa). The underlying mechanism for the observed mobility shift (smaller than calculated) is unclear but appears consistent with tetraspanin proteins, as previously reported [53,54]. To determine whether the mobility shift was specific to the GFP-CD63 construct, we engineered a CD63-fused HIV protease precursor (CD63-M1-PR^HIV^-HA) for comparison. Interestingly, this CD63 fusion exhibited a detection pattern (lane 8 of the upper panel) distinct from endogenous CD63 and GFP-CD63: both a smear and a more defined CD63-M1 band were detected, which converted to dCD63-M1 upon PNGase F treatment (lane 9). Both CD63-M1 and dCD63-M1 ran faster than their anticipated sizes. Nonetheless, our data confirmed the effective removal of CD63 N-glycosylation by PNGase F treatment, which facilitated Western blotting detection.

We then compared the N−terminal processing of the CD63-fused M^pro^ precursors with either Q or E at the N−terminal P1 position. A lysate aliquot that did not go through the deglycosylation procedure was included (lanes 12, 15, and 18) as an input control because protein degradation in mock-treated lysates was noticed even in the presence of a protease inhibitor cocktail and without the PNGase F enzyme. For example, the deglycosylation procedure reduced M^pro^-HA detection by about ~50% (lane 12 vs. 13 and 14 in the lower panel). This was even more prominent with the full-length precursors (lane 15 vs. 16 in the lower panel, and lane 18 vs. 19 in both panels), showing >80% reduction in detection. Nevertheless, our results demonstrated an almost complete N-terminal processing by the Q precursor in the CD63 fusion context. The E precursor showed ~80% N-terminal processing efficiency determined through the HA channel (Figure 4B, bottom panel, lane 17), confirming that the mutation of Q to E reduced precursor activity but did not abolish N-terminal processing. Taken together, unlike sGST fusion tag, CD63 fusion had minimal impact on N-terminal processing.

Using mouse hepatitis virus (MHV) A59 as a coronavirus model, it was previously demonstrated that N-linked glycosylation of MHV Nsp4 is involved in membrane modifications through mitochondria [53,55]. This is distinct from CD63, which primarily undergoes exocytosis and late endocytosis. We therefore constructed and examined precursors fused to Nsp4, the native upstream viral protein. To facilitate detection, a 4xFlag tandem tag was inserted 15 amino acids upstream of the Nsp4/Nsp5 (M^pro^) cleavage site (Figure 4A, the right panel). Interestingly, we detected multiple and smearing Flag-reactive bands that were unresponsive to PNGase F treatment (Figure 4B, lanes 25 through 30), suggesting unconventional glycosylation of SARS-CoV-2 Nsp4 in transfected HEK 293T cells. This unusual glycosylation impeded any meaningful quantification as with CD63-fused precursors, necessitating further investigations beyond the scope of this study. Nevertheless, precursors with either Q or E at the P1 position both appeared autoprocessing-competent, releasing approximately equal amounts of M^pro^-HA (Figure 4B, lanes 25 and 26 vs. lanes 27 and 28), which is absent in the C145G mutant (lanes 29 and 30). This further confirmed that while Q being at P1 might be preferred, it is not required. We also detected some differences between precursor Q_E and E_E in the Flag channel. Precursor E_E produced extra Flag-reactive bands running at ~45 kDa, which were absent in the Q_E lysates. These bands were reminiscent of those detected from cells expressing SARS-CoV Nsp4-HA in a previous report, which showed multiple SARS-CoV Nsp4 bands caused by partial (incomplete) signal peptide processing at the N-terminus and atypical N-glycosylation [53]. These bands were also detected in the C145G lysates, suggesting their production was independent of conventional M^pro^ catalysis. Collectively, our results implied additional complex regulation of precursor autoprocessing by the N-terminal P1 residues when fused to its native upstream Nsp4.

### 3.5. Diverse and Distinct Susceptibilities of Precursor and Released M^pro^ to GC376

To further assess whether mutations at the N-terminal P1 position simply alter the substrate sequence or have additional impacts, we examined the effects of compound GC376, a pre-clinical broad-spectrum drug that inhibits M^pro^ activities by forming a covalent adduct to Cys145 at the active site [31]. Similar to nirmatrelvir, an orally active M^pro^ inhibitor developed by Pfizer and marketed as Paxlovid (nirmatrelvir/ritonavir combination), GC376 has a rigid pyrrolidone mimicking the glutamine side chain in the P1 position for competitive binding. GC376 can target at least two forms of M^pro^: the precursor and the released enzyme. Binding to the precursor active site inhibits N-terminal processing, leading to the accumulation of the full-length (FL) precursor, while binding to the released M^pro^ suppresses its activities. If the precursor and the released M^pro^ are catalytically identical, similar EC_50_ values should be observed; otherwise, different susceptibilities will be detected.

GST-fused constructs were utilized for ease of detection via Western blotting. By quantifying the band intensity of the FL precursor normalized to GAPDH as a function of the GC376 concentration, we found that GC376 suppressed N-terminal processing of the Q (wild-type) precursor with an apparent EC_50_ of ~2 µM (Figure 5A,B, grey line). Meanwhile, the released M^pro^ exhibited a bell-shaped detection profile (Figure 5B, magenta line). Low detection in the absence of the inhibitor was consistent with its known self-degradation property [17,18,19,21], which was suppressed by GC376, leading to increased M^pro^ detection with an apparent EC_50_ of ~0.1 µM (left half of the curve). At higher GC376 concentrations, decreased M^pro^ detection (right half of the curve) inversely correlated with increased FL precursor detection due to suppressed N-terminal processing. The apparent EC_50_ for the released M^pro^ (~0.1 µM) matched well with the in vitro IC_50_ values determined with recombinant M^pro^ [31,56,57]. Collectively, our data revealed that the precursor and its released M^pro^ are enzymatically different, as precursor-mediated N-terminal processing was ~20-fold more resistant to GC376 inhibition than the released M^pro^ in the same transfected cells. This difference in GC376 susceptibility implies differential catalysis between the precursor and the mature M^pro^, indicating the potential of targeting precursor-mediated N-terminal processing for novel antiviral development.

GST−fused E precursor and the released M^pro^ responded to GC376 treatment differently from the corresponding Q precursor in several aspects (Figure 5C,D). The E precursor was less effective at N−terminal processing, resulting in detection of the FL precursor even without GC376. Additionally, the N−terminal processing was highly sensitive to GC376 inhibition, with an apparent EC_50_ of ~20 nM. Furthermore, the released M^pro^ was resistant to self-degradation, as its detection remained constant with or without GC376 until N-terminal processing was suppressed. Therefore, our data suggest that the observed differences in the self-degradation properties of the released M^pro^ originate from the precursor autoprocessing step, where P1 residue variations modulate both precursor catalysis and GC376 susceptibility. We also compared GC376 and nirmatrelvir using the GST-fused Q precursor as a test model (Figure 5E). The results of our side-by-side analysis further confirmed that precursors are less sensitive than the released M^pro^ to drug suppression (solid lines right-shifted of dashed lines). Both drugs demonstrated similar efficacies, with nirmatrelvir being slightly more potent than GC376. We thus focused on GC376 in the subsequent experiments for its cheaper cost.

We repeated the GC376 analysis in the context of Nsp4 fusion for its biological relevance. Only the released M^pro^ was detectable because Nsp4-fused proteins cannot be quantified by WB due to atypical glycosylation (Figure 4). Once again, the M^pro^ released from the Q precursor was prone to self-degradation, which could be suppressed by GC376 (Figure 5F, solid line), whereas the M^pro^ released from the E precursor was resistant to self-degradation (Figure 5F, dashed line). Therefore, our data confirmed that precursor catalysis is flexible and subject to regulation by P1 residue alterations, resulting in the liberation of M^pro^ with distinct properties (self-degradation prone vs. resistant). The results of a parallel comparison also revealed a modulation of GC376 susceptibility by different fusion tags (Figure 5F). The M^pro^ released from the Nsp4-fused Q precursor was more resistant to GC376 than that from the GST-fused one (right-shifted by a few folds). Additionally, the M^pro^ released from the Nsp4-fused E precursor was much more resistant to GC376 than that from its GST-fused counterpart (right-shifted by ~100-fold). Taken together, our results demonstrate complex regulation of precursor catalysis by the N-terminal P1 residue and various upstream fusion tags, which also dictates the properties of the released M^pro^ that is enzymatically different from the precursor. These differences validated the precursor as another promising target for antiviral development [58].

### 3.6. Assay Adaptation and Validation for Dual High-Throughput Screening

To enable high-throughput screening (HTS) for novel N-terminal processing inhibitors, we employed AlphaLISA, a label-free and wash-free platform [59,60,61,62], for sensitive quantification of the FL precursor, which is reversely correlated with the N-terminal processing efficiency (Figure 5A,B). Our previous HTS endeavors have identified glutathione-coated donor and anti-FLAG-coated acceptor beads as an optimal pair [46]. Accordingly, we engineered two versions of fusion constructs for evaluation. The Nsp4-4F-M^pro^-GST is nearly identical to Nsp4-4xFlag-M^pro^-HA (Figure 4A, right panel) except that the HA tag is replaced by the GST to allow for AlphaLISA quantification. The 4F-M^pro^-GST construct has the Nsp4 portion upstream of the 4xFlag insertion removed and thus is expected to be mainly cytoplasm-associated like the GST-fused precursors.

In the process of our assay development, we serendipitously identified another reporter for the released M^pro^ activity. As a routine practice, we normally include a trace amount of GFP-encoding plasmid (<3% of total input DNA) to monitor transfection efficiency. When co-transfected with the wild-type precursor in the absence of an inhibitor, we consistently observed reduced levels of GFP detection. Furthermore, GFP detection increased as the concentration of GC376 increased in a dose-dependent manner (Figure 5A, lower panel). Furthermore, consistently high GFP signals were detected in cells expressing the C145G mutant, which is deficient at N-terminal processing and releases no M^pro^-Flag (Figure 5A). These results suggest that the released M^pro^-Flag is likely responsible for the promiscuous degradation of GFP. This is not entirely surprising, as previous reports have indicated that recombinant SARS-CoV-2 M^pro^ could process many non-viral host cell proteins, including GFP [49]. Nonetheless, this provided a convenient fluorescence reporter for the released M^pro^-Flag activity: low fluorescence signals correlated with high M^pro^-Flag activity, and vice versa.

The minimal spectral overlap between GFP and AlphaLISA signals (Figure 6A) allowed for simultaneous monitoring of the released M^pro^ activity (via GFP) and the N-terminal processing efficiency (via AlphaLISA) in the transfected cells treated with or without inhibitors. The M^pro^-GST liberated from both fusion precursors (4F-M^pro^-GST and Nsp4-4F-M^pro^-GST) demonstrated significant sensitivity to GC376 inhibition, with apparent EC_50_ values in the low-hundred nM range (Figure 6B,C, green). This sensitivity aligns with the WB detection profiles of the released M^pro^ from GST- or Nsp4-fused Q precursors (Figure 5F). These results validated the GFP signal as a proxy for the released M^pro^-GST activity. AlphaLISA analysis showed suppression of N-terminal processing by GC376 in the low µM range (Figure 6B,C, red). Despite minor differences (<2 µM for Nsp4-4F-fused and ~3 µM for 4F-fused precursors), the AlphaLISA results were consistent with the WB results, showing an EC_50_ value of ~2 µM for the GST-fused Q precursor (Figure 5B). Our data confirmed that the precursor and the released M^pro^ are enzymatically distinct, showing that the released M^pro^ was more responsive (left-shifted) to GC376 inhibition than precursor-mediated N-terminal processing (Figure 6B,C). Collectively, these assessments validated the screening platform for simultaneously evaluating drug effects on both M^pro^ activity and precursor-mediated N-terminal processing in transfected cells.

To further evaluate this platform, we screened a small-protease-inhibitor library (Apex #1035, 130 compounds) at 20 uM in duplicates in a 384-well plate against the precursor Nsp4-4F-M^pro^-GST (Figure 6D). Boceprevir, calpain inhibitor II, Z-FA-FMK, and LY450139 showed higher GFP signals than those of the DMSO controls, suggesting that they suppressed GFP degradation mediated by the released M^pro^-GST. Indeed, these inhibitors were previously identified as positive hits by FRET-based in vitro enzymatic assays [57,63] or a cell-based assay expressing mature M^pro^ [64]. Therefore, our re-identification of these inhibitors validated our GFP detection as a reliable readout for identifying inhibitors of the released M^pro^.

To further evaluate assay performance, we tested a few compounds at 25 µM each twelve times (to obtain HTS statistics) against these two constructs (Figure 7). Our data illustrated that atazanavir and lopinavir (two known HIV-1 protease inhibitors) and E-64 (a cysteine protease inhibitor) showed background levels (DMSO control) of both GFP and AlphaLISA signals, suggesting no inhibitory effects on either the released M^pro^-GST or precursor-mediated N-terminal processing. GC376 positive controls suppressed both GFP degradation and N-terminal processing at 25 µM, as indicated by the GFP and AlphaLISA signals above the baseline, which were consistent with the Western blotting WB results (Figure 5E) and dose–response analyses (Figure 6B,C). At 25 µM, Z-FA-FMK, boceprevir, and calpain inhibitor II consistently suppressed the GFP degradation caused by the released M^pro^-GST, as indicated by increased GFP signals (Figure 7, upper panel), confirming the utility of GFP detection. However, these inhibitors showed no detectable impact on precursor-mediated N-terminal processing, as no increased detection of the FL precursor was observed, supporting the idea that precursors are less responsive to these M^pro^ inhibitors. We also noticed some differences between the two constructs and speculated that the Nsp4 fusion plus an internal insertion of 4xFlag were likely causes of the observed differences. Consequently, the precursor 4F-M^pro^-GST appeared to be a better candidate for HTS, demonstrating S/N ≥ 5.0, Z’ ≥ 0.3, and CV ≤ 15% for GFP detection and S/N ≥ 13.0, Z’ ≥ 0.6, and CV ≤ 12% for AlphaLISA detection, all of which are strong statistical values for a cell-based HTS assay. Given that precursors are subject to complex regulation, precursor Nsp4-4F-M^pro^-GST can be used for hit validation as an orthogonal assay if needed. Taken together, we have optimized and validated an HTS platform with dual detection of precursor-mediated N-terminal processing and GFP degradation by the released M^pro^ through AlphaLISA and GFP reporter signals, respectively. Therefore, we are poised to conduct HTS campaigns for the identification of chemicals that inhibit these two enzymes (precursor and the released M^pro^) via various mechanisms.

## 4. Discussion

This report describes an assay platform for studying SARS-CoV-2 M^pro^ precursor autoprocessing in transfected mammalian cells. Our results reveal several intriguing aspects of M^pro^ precursor catalysis, providing molecular insights into the regulation of precursor autoprocessing. We also developed and validated an HTS-compatible platform for drug discovery, simultaneously targeting SARS-CoV-2 M^pro^ precursor autoprocessing and the released mature M^pro^.

Distinct proteolysis kinetics at N- or C-terminal processing sites—To liberate free mature M^pro^ from polyproteins 1a and 1ab, the precursors must autoprocess both N- and C-terminal cleavage sites, Nsp4↓M^pro^ and M^pro^↓Nsp6 (Figure 1A). Our results show that blocking C-terminal processing had minimal impact on N-terminal processing. However, mutating Q to E at the N-terminal P1 position reduced both N- and C-terminal processing (Figure 1B, lane 15). This suggests that the P1 residue modulates precursor catalysis in the context of N-terminal extensions but has less influence on C-terminal extensions. Similar asymmetrical effects have been documented for HIV-1 protease autoprocessing, where N-terminal processing is critical for mature HIV protease liberation [65,66,67] and C-terminal processing is less influential [68,69]. These findings indicate another regulatory dimension for viral proteases synthesized as part of a polyprotein and subsequently liberated through precursor autoprocessing. This speculation explains well why different upstream fusion tags have diverse impacts on SARS-CoV-2 N-terminal processing (Figure 3 and Figure 4) and HIV-1 protease autoprocessing [45,46,47,48,70,71,72,73], aligning with previous reports showing that N-terminal extensions impact precursor dimerization and catalysis more than C-terminal extensions [12,26,42,74].

SARS-CoV-2 precursor catalysis modulated by upstream fusion tags—We investigated several fusion tags, categorized into two groups, based on subcellular localization. Fusion precursors downstream of the GST and 4xFlag tags are expected to be mainly cytosol-associated, while those downstream of sGST, CD63, and Nsp4 are anticipated to be near different membranes. Our results revealed approximately complete N-terminal autoprocessing of the wild-type precursor when fused to the cytosolic tags (Figure 1C). However, when fused to sGST, the N-terminal autoprocessing efficiency was reduced to ~40% (Figure 3C). When fused to CD63 or Nsp4, the wild-type precursor appeared competent at N-terminal processing, although complex glycosylation of these tags impeded reliable quantification (Figure 4). Nonetheless, these analyses demonstrate the flexibility of precursor catalysis in response to modulation by upstream elements in the context of polyproteins.

SARS-CoV-2 precursor catalysis modulated by the N-terminal P1 residue—Residue Q at the P1 position is highly conserved among various substrate sequences of coronavirus polyproteins [10,11,12] and is believed to be crucial for effective proteolysis. This is indeed true for M^pro^ precursor autoprocessing at the C-terminus of SARS-CoV-2 (Figure 1B, lane 13) and the N-terminus of MERS (Figure 1B, lane 9). However, Q at the N-terminal P1 position plays multiple roles in modulating SARS-CoV-2 M^pro^ precursor catalysis. As a substrate residue, while Q is preferred [11,18], it is not strictly required, as many amino acid variations can support N-terminal processing (Figure 2). Meanwhile, the mutation of Q to E reduced precursor processing efficiency at both the N- and C-terminal sites (Figure 1, lane 15). This differs from previous reports indicating that substrate peptides with the P1 residue mutated to D, E, or K were not cleavable by purified SARS-CoV M^pro^ [75] and that fusion precursors with an N-terminal extension carrying a mutated P1 residue (E or N) were deficient in N-terminal autoprocessing when expressed in and purified from *E. coli* or produced by an *E. coli* cell-free protein synthesis system [12,42]. Therefore, our results suggest that fusion precursors expressed in transfected mammalian cells are not catalytically identical to those made in *E. coli* or the cell-free system, implying a conformational plasticity of M^pro^ precursors in response to different cellular (prokaryotic vs. eukaryotic) environments and upstream contexts.

Additionally, the Q-to-E mutation at the N-terminal P1 position altered the enzymatic properties of the released M^pro^. The M^pro^ released from wild-type Q precursors was prone to self-degradation and responsive to GC376 suppression, while M^pro^ released from E precursors was resistant to self-degradation and unresponsive to inhibitor suppression (Figure 5). This was reproducible in all four fusion contexts tested, validating the role of the N-terminal P1 residue in modulating precursor catalysis. Such conformational plasticity presents challenges to structure-based antiviral development, warranting alternative approaches that account for this variability.

Precursors and the released M^pro^ are enzymatically different—We previously reported that mature HIV protease is enzymatically different from its precursors, responding differentially to protease inhibitors designed to specifically target the catalytic site of the mature protease [44,47]. These findings validated HIV protease precursors as novel targets for antiretroviral development [45,46,73] to combat drug resistance. Similarly, using a cell-based assay and GC376, a preclinical M^pro^ inhibitor, we detected catalytic differences between SARS-CoV-2 M^pro^ precursors and the released mature M^pro^ (Figure 6). Notably, these precursors were less responsive than the released M^pro^ to suppression by several tested inhibitors, suggesting catalysis differences between these two forms of protease, which is consistent with reports showing modulation of precursor activities by upstream sequences and even inhibitor binding [13,41,76]. Since previous anti-COVID drug development has primarily focused on mature M^pro^ [33,34], our findings present the precursor as an alternative target [58] for the discovery of inhibitors with mechanisms distinct from existing M^pro^ inhibitors. Our development and validation of the dual-targeting assay platform provides a useful tool to identify these promising drugs through HTS.

## Figures and Tables

**Figure 1 viruses-16-01218-f001:**
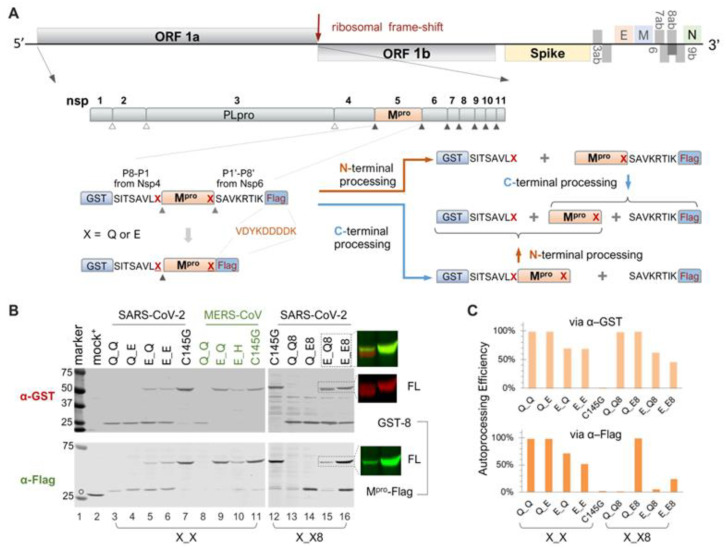
Autoprocessing of M^pro^ fusion precursors. (**A**) Schematics of SARS-CoV-2 genome and fusion precursors, and the anticipated autoprocessing intermediates and final products. Open triangles indicate sites processed by papain-like protease (PL^pro^), while solid triangles were processed by 3C-like protease, or Nsp5, or the main protease (M^pro^). The precursor X_X8 contains both the N- and C-terminal cleavage sites (P8-P1↓P1′-P8′), whereas precursor X_X has the M^pro^ C-terminus directly fused to a Flag tag. The letter “X” denotes P1 residues. (**B**) Autoprocessing of fusion precursors in transfected HEK293T cells with orange arrows denoting N-terminal processing and blue arrows C-terminal processing. Post-nuclear lysates were resolved using SDS-PAGE and simultaneously probed with rabbit anti-GST and mouse anti-Flag antibodies for dual visualization with an Odyssey imaging unit. Also included are the FL precursor in the last two lanes, through individual channels and a merged image. The open circle denotes GFP-Flag, included in the mock-transfected lysates as a reference. A bracket line connects the two anticipated autoprocessing products. (**C**) Quantification of autoprocessing efficiency. The band intensity of the autoprocessing product, expressed as a percentage of the total antibody-reactive bands (product + FL) through either the GST or Flag channel, was determined to represent autoprocessing efficiency. Data are representative of two independent experiments.

**Figure 2 viruses-16-01218-f002:**
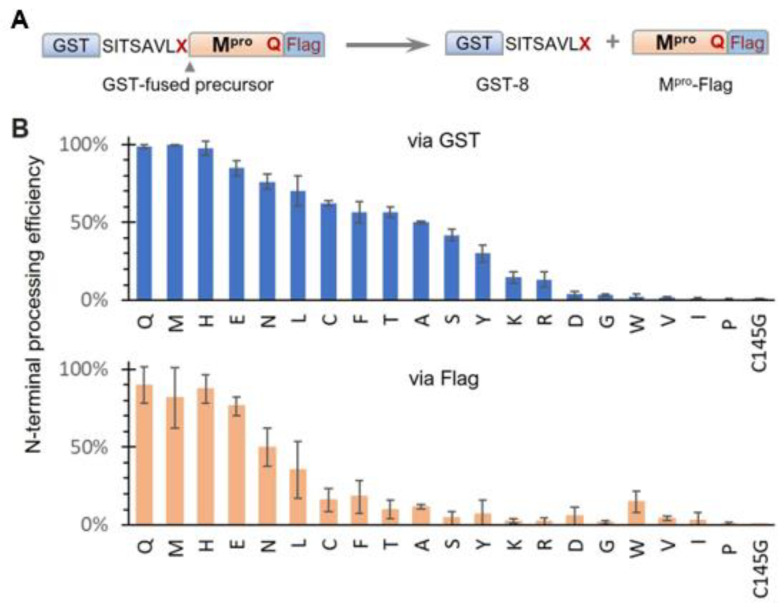
Effects of N-terminal P1 residue variations on precursor N-terminal autoprocessing. (**A**) Schematic of GST-fused precursor construct and its mediated N-terminal autoprocessing. (**B**) Autoprocessing efficiencies calculated via GST-reactive or Flag-reactive bands. Bars represent standard deviations of data from five independent experiments.

**Figure 3 viruses-16-01218-f003:**
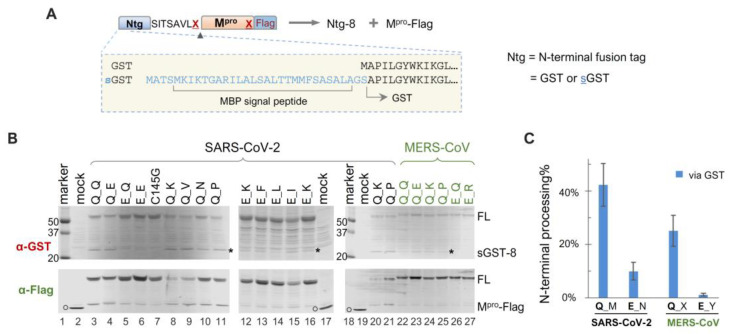
Autoprocessing analyses of sGST-fused M^pro^ precursors. (**A**) Schematics of sGST-fused precursors highlighting the difference from GST-fused precursors. Construct nomenclature is described as in Figure 1. (**B**) Precursor autoprocessing in transfected HEK293T cells. Post-nucleus lysates were resolved through SDS-PAGE and simultaneously probed with rabbit anti-GST and mouse anti-Flag for dual visualization via an Odyssey imaging unit. The asterisks denote sGST-8, an expected autoprocessing product; the open circles denote GFP-Flag that was included in the mock-transfected lysates serving as a size reference. Data are representative of three independent experiments. (**C**), Quantification of autoprocessing efficiency by GST signal. Bars denote standard deviations; C-terminal P1 residue variations are denoted by different letters (M = Q, E, N, K, V, P, V; N = Q, I, K, F, L; X = Q, D, K, L, A, P, V; Y= Q, E, T, R, K, H).

**Figure 4 viruses-16-01218-f004:**
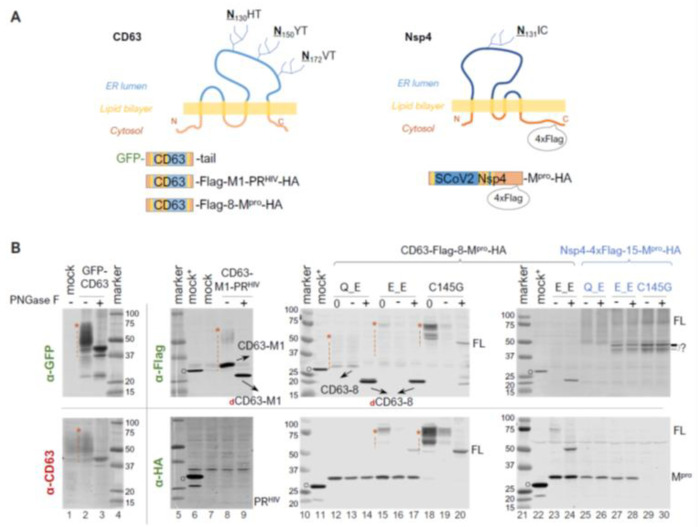
N-terminal autoprocessing of M^pro^ precursors fused to membrane-associated CD63 and SARS-CoV-2 Nsp4. (**A**) Schematics of fusion constructs with anticipated membrane topology. Three N glycosylation sites in CD63 are indicted. While SARS-CoV-2 Nsp4 lacks conventional N-glycosylation sequences (NXS/T), it contains an atypical glycosylation motif (N_131_IC). (**B**). Fusion precursor autoprocessing in transfected HEK293T cells. Post-nucleus lysates were resolved through SDS-PAGE, followed by probing with the indicated antibodies and visualized using an Odyssey imaging unit. In the figure, “0” denotes that a lysate input aliquot did not go undergo deglycosylation. Open circles indicate GFP-Flag and/or GFP-HA that were included in mock-transfected lysates serving as a size reference. The red asterisk followed by a vertical dash line marks glycosylated CD63-containing proteins. Data are representative of two independent experiments.

**Figure 5 viruses-16-01218-f005:**
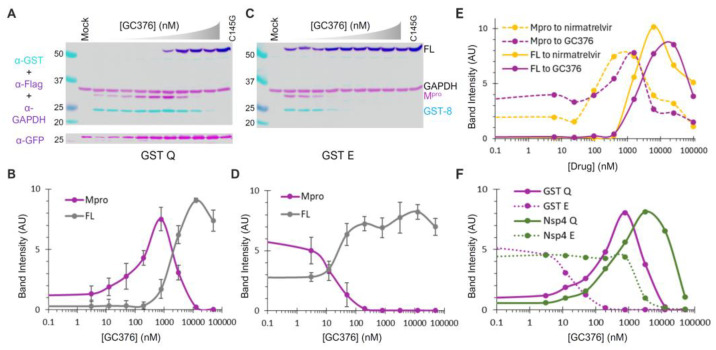
Differential GC376 susceptibilities of fusion precursors and the released M^pro^. (**A**–**D**) GC376 response profiles of GST-fused precursor Q or E. HEK 293T cells were transfected with the indicated constructs and treated with increasing concentrations of GC376. Post-nuclear lysates were resolved through SDS-PAGE, followed by probing with the indicated antibodies and visualization using an Odyssey imaging unit (**top panels**). Band intensity, normalized to GAPDH, was plotted to GC376 concentration (**bottom graphs**), with error bars representing standard deviations of three to five independent experiments. Error bars in (**B**,**D**) denote standard deviations from three to five independent experiments. (**E**) Response profiles of the FL precursor and M^pro^ released from GST−fused precursors to GC376 and nirmatrelvir. (**F**) A comparative analysis of M^pro^s released from the indicated precursor. Standard deviation error bars for panels (**E**,**F**) are available but not shown for clarity.

**Figure 6 viruses-16-01218-f006:**
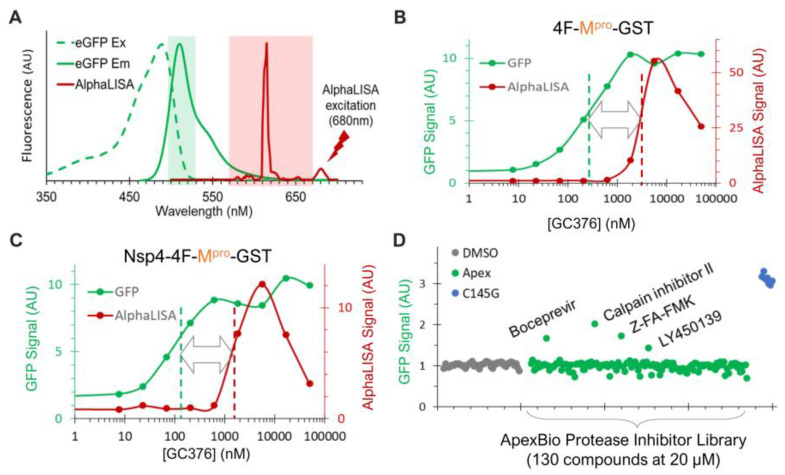
Dual detection of precursor-mediated N-terminal processing and GFP degradation by the released M^pro^. (**A**) Dual detection is based on minimal spectral overlap between AlphaLISA (red) and GFP (green) signal. (**B**,**C**) HEK293T cells transfected with the indicated constructs and 1% GFP plasmid were treated with increasing concentrations of GC376 for 24 h. After in situ cell lysis and mixing with AlphaLISA reagents, crude lysates were subjected to GFP detection (green) followed by AlphaLISA quantification (red). (**D**) Pilot screen of ApexBio protease inhibitors. Only GFP signals are shown; AlphaLISA signals remained at basal levels and are not displayed.

**Figure 7 viruses-16-01218-f007:**
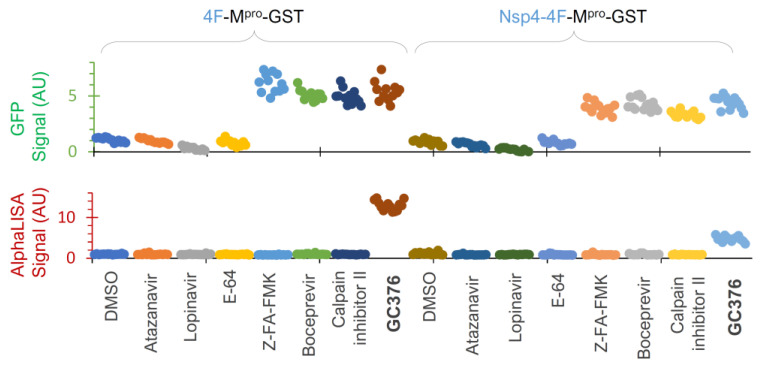
Assay validation. HEK 293T cells were transfected with two indicated constructs plus 3% GFP plasmid and treated with the indicated compounds at 25 µM for 24 h in a 384-well plate, followed by dual detection of GFP (upper) and AlphaLISA (lower) signals. Different colors represent different compounds.

## Data Availability

The original contributions presented in the study are included in the article, further inquiries can be directed to the corresponding author. The raw data supporting the conclusions of this article will be made available by the authors on request. The plasmids used in this study are available with a standard material transfer agreement.

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
