# Peer review of "Assay Development and Validation for Innovative Antiviral Development Targeting the N-Terminal Autoprocessing of SARS-CoV-2 Main Protease Precursors"

_viruses, 2024, doi:10.3390/v16081218_

Round 1
Reviewer 1 Report
Comments and Suggestions for Authors
To understand how SARS-CoV-2 is replicated, it is important to investigate how the long polypeptide ORF1ab is processed. This paper provides new insights into how this arises in relation to Mpro, which is a key player for this. This is an enzyme that is also a target for therapeutic drugs, so this is an important finding in that respect.
I would like to make two requests.
There are more papers on the function of Mpro. These should be introduced in the introduction, as they are, after all, the importance of this paper.
There is a lack of information on plasmid construction: the Mpro is part of a long coding frame, lacking methionine at the N-terminus. You need to describe in more detail what construction you used, so we can understand the experiment and reproduce it.
The following is not relevant to this review.
It should be emphasised that Mpro is a particularly mutation-prone location in ORF1ab, which is comparatively less mutated. For example, the attached figure shows
https://www.mdpi.com/2673-8112/4/4/38
The figure is modified from Fig. 2 of and shows that the mutations are particularly high in the area including the processing site. This would suggest that the precursor problem pointed out in this paper is a possible problem. You can use this figure as you wish or ignore it if you wish.
This is just a layman's idea, but I was curious about the difference in processing efficiency between proteins with transmembrane domains such as NSP4 and GSTs at the N-terminus. It is possible that the N-terminal processing is not an auto catalysis: the signal peptide is cleaved by a protease in the ER. Is it possible that this is what is doing the processing of the N-terminus? I would be personally happy if you could DISCUSS this.

Author Response
This reviewer commented that our manuscript provides new insight into how the long polypeptide ORF1ab is processed in relation to Mpro. Two requested revision points are addressed as follow:
- “There are more papers on the function of Mpro. These should be introduced in the introduction, as they are, after all, the importance of this paper. We added several related references to the Introduction, which are also highlighted in the submitted revised version.
- “There is a lack of information on plasmid construction: the Mpro is part of a long coding frame, lacking methionine at the N-terminus. You need to describe in more detail what construction you used, so we can understand the experiment and reproduce it.” In this report, we indeed engineered and examined a collection of mammalian expression plasmids. Instead of describing them together in the Materials and Methods section, we introduced them in each corresponding result section to provide context-related information. We acknowledge that this approach might have diluted information on plasmid construction. Accordingly, we added a sentence to better guide reader’s attention to find the related information (highlighted in lines 86-8).
Reviewer 2 Report
Comments and Suggestions for Authors
The authors submitted an interesting manuscript entitled "Assay Development and Validation for Innovative Antiviral Development Targeting the N-terminal Autoprocessing of SARS-CoV-2 Main Protease Precursors" for review.
The authors' research results revealed several aspects of precursor catalysis, showing that the precursors are enzymatically different from the released Mpro. This observation offers alternative targets for antiviral drug development. Additionally, the authors developed a screening platform to enable functional screening for compounds targeting both the precursor and released Mpro through different mechanisms of action.
Therefore, due to the importance of the content in the development of new, more effective antiviral drugs, detailed preparation of the experimental part and quite good discussion, this manuscript can be published in the journal Viruses after correcting several shortcomings mentioned below:
On line 31 is … ([1-5]) … , but should be better … (1–5) … . Comment: Currently, a medium "–" sign is used between numbers. Similar mistake to correction is at lines: 45, 47 (twice), 62 (twice), 78, 103, 104, 107, 110, 120, 145, 149, 151, 190, 226, 243, 414, 419, 466, 533, 566, 571 (twice), 587, 645 (ref. [1]), 648 (ref. [2]), 650 (ref. [3]), 657 [5], 659 [6], 660 [7], 661 [8], 664 [9], 667 [10], 669 [11], 672 [12], 674 [13], 676 [14], 682 [16], 688 [18], 690 [19], 693 [20], 695 [21], 698 [22], 702 [24], 709 [27], 716 [30], 718 [31], 739 [40], 742 [41], 744 [42], 753 [45], 757 [47], 760 [48], 763 [49], 765 [50], 768 [51], 774 [54], 776 [55], 778 [56], 781 [57], 783 [58], 790 [61], 793 [62], and 795 [63].
At lines 117 and 142 are … -20°C … , … 100°C … , respectively, but at line 118 is … 95 ÌŠ C … . Comments: Please use standards degree “ ° “ and mathematic subtraction “ – “ marks.
At line 119 is: … 3000xg … , but maybe should be … 3000×g … , or better … 3000 × g … . Comment: The mathematical multiplication sign must be used. There is a similar error to be corrected in line e.g. 139. The scientific style of the manuscript is very desirable.
The text from line 210 should be directly below line 209. Comment: Probably an editorial error in the 'setting' of the manuscript consisting in inserting Figure 1 above the last line of the paragraph, which seriously hinders the MDPI reader from studying the content of the manuscript.
As above, lines 363–370 should be placed directly below line 362. Comment: Please, if possible, 'do not break' the content of the paragraphs.
There is one dot too many on line 557 at the end of the caption for figure 7.
At line 678 (ref. [15]) is … Science 2020. … , but please complete it. Comments: Please add the article number or page range and volume number.
At line 706 (ref. [26]) is … 2024, 16, (2). … , but please finish this text. Comment: Please add the article number.
At line 711 (ref. [28]) is … 2020, 21, (9). … , but please finish this text. Comment: Please add the article number.
At line 714 (ref. [29]) is … 2023, 15, (9) … , but please finish this text. Comment: Please add the article number.
At line 726 (ref. [35]) is … Sci Rep 2019. … , but please finish this text. Comments: Please add the volume number and article number.
At line 737 (ref. [39]) is … 2020, 6, (50). … , but please finish the text. Comment: please add article number or page range.
At line 754 (ref. [46]) is … 2021, 13, (10). … , but please finish this text. Comment: Please add the article number.
At line 770 (ref. [52]) is … 2021, 13, (2). … , but please finish this text. Comment: Please add the article number.
At line 772 (ref. [53]) is … Nat.Struct.Biol. 1999, 6, 868. … . Comment: Please standardize the abbreviation style of the magazine name.
At line 754 (ref. [46]) is … 2021, 13, (10). … , but please finish this text. Comment: please add article number.
The introduction should include references to current source literature in the discussed field or related fields, along with a short critical discussion of the state of knowledge. Only about 10 percent of the current literature and 20% of the source literature from the last five years are not numbers that convince the reader to read the manuscript more deeply, even though the research presented concerns contemporary problems of humanity.
The introduction of a Conclusions chapter may contribute to a significant improvement in the quality of the manuscript, as the abstract is not formally part of the manuscript.
Author Response
Responses to review #2:
This reviewer highlighted the importance of this manuscript and recommended it for publication after correcting several shortcomings that are mainly related to reference citation and editorial formatting:
- “On line 31 is … ([1-5]) … , but should be better … (1–5) … . Comment: Currently, a medium "–" sign is used between numbers. Similar mistake to correction is at lines: 45, 47 (twice), 62 (twice), 78, 103, 104, 107, 110, 120, 145, 149, 151, 190, 226, 243, 414, 419, 466, 533, 566, 571 (twice), 587, 645 (ref. [1]), 648 (ref. [2]), 650 (ref. [3]), 657 [5], 659 [6], 660 [7], 661 [8], 664 [9], 667 [10], 669 [11], 672 [12], 674 [13], 676 [14], 682 [16], 688 [18], 690 [19], 693 [20], 695 [21], 698 [22], 702 [24], 709 [27], 716 [30], 718 [31], 739 [40], 742 [41], 744 [42], 753 [45], 757 [47], 760 [48], 763 [49], 765 [50], 768 [51], 774 [54], 776 [55], 778 [56], 781 [57], 783 [58], 790 [61], 793 [62], and 795 [63].”? This reference citation format was automatically populated by EndNote 20 software following the “Viruses” style. We’d be glad to change them if more specific instructions are provided.
- “At lines 117 and 142 are … -20°C … , … 100°C … , respectively, but at line 118 is … 95 ÌŠ C … . Comments: Please use standards degree “ ° “ and mathematic subtraction “ – “ marks.” Corrected as suggested.
- “At line 119 is: … 3000xg … , but maybe should be … 3000×g … , or better … 3000 × g … . Comment: The mathematical multiplication sign must be used. There is a similar error to be corrected in line e.g. 139. The scientific style of the manuscript is very desirable.” Corrected as suggested.
- “The text from line 210 should be directly below line 209. Comment: Probably an editorial error in the 'setting' of the manuscript consisting in inserting Figure 1 above the last line of the paragraph, which seriously hinders the MDPI reader from studying the content of the manuscript.” Corrected as suggested.
- “As above, lines 363–370 should be placed directly below line 362. Comment: Please, if possible, 'do not break' the content of the paragraphs.” Corrected as suggested.
- “There is one dot too many on line 557 at the end of the caption for figure 7.” Corrected as suggested.
- “At line 678 (ref. [15]) is … Science 2020. … , but please complete it. Comments: Please add the article number or page range and volume number.
At line 706 (ref. [26]) is … 2024, 16, (2). … , but please finish this text. Comment: Please add the article number.
At line 711 (ref. [28]) is … 2020, 21, (9). … , but please finish this text. Comment: Please add the article number.
At line 714 (ref. [29]) is … 2023, 15, (9) … , but please finish this text. Comment: Please add the article number.
At line 726 (ref. [35]) is … Sci Rep 2019. … , but please finish this text. Comments: Please add the volume number and article number.
At line 737 (ref. [39]) is … 2020, 6, (50). … , but please finish the text. Comment: please add article number or page range.
At line 754 (ref. [46]) is … 2021, 13, (10). … , but please finish this text. Comment: Please add the article number.
At line 770 (ref. [52]) is … 2021, 13, (2). … , but please finish this text. Comment: Please add the article number.
At line 772 (ref. [53]) is … Nat.Struct.Biol. 1999, 6, 868. … . Comment: Please standardize the abbreviation style of the magazine name.” Corrected as suggested.
- “The introduction should include references to current source literature in the discussed field or related fields, along with a short critical discussion of the state of knowledge.” We added more references and a short summary of the state of knowledge as suggested (lines 51-3).
- “The introduction of a Conclusions chapter may contribute to a significant improvement in the quality of the manuscript, as the abstract is not formally part of the manuscript.” We added more references and a short summary of the state of knowledge as suggested (lines 72-9).
Reviewer 3 Report
Comments and Suggestions for Authors
Comments
This paper clarified the effects of adjacent amino acids in the SARS-CoV-2 main protease (Mpro) precursor on the process by which it is autoproteolyzed to generate mature Mpro.
It is significant finding that mutations in the adjacent amino acids in the Mpro precursor cause differences in self-processing and affect virus production, and the paper is worthy of publication.
However, a few points need to be addressed.
The experiments were only conducted in experimental cells. Therefore, do such mutations actually occur in the natural world?
If virus production is hindered when such mutations occur, virus production will eventually cease. Therefore, it may be meaningless to link mutations in the adjacent amino acids in the Mpro precursor to the development of antiviral drugs.
Does the development of antiviral drugs refer to drugs that cause mutations in the adjacent amino acids? Molnupiravir was reported to have an antiviral effect by changing the amino acid sequence, and there is a powerful paper showing that such an approach is effective.
If Mpro with adjacent mutations amino acids is synthesized and incorporated into the virus, and if the resulting protein is functional, it will create a completely new mutant virus, which is very interesting from a clinical level.
If there are similar examples for other viruses, it would be good to introduce them.
If such a mechanism can produce viral resistance, clarification is awaited from a molecular perspective regarding the structural differences the adjacent amino acids produce in wild-type Mpro.
Author Response
This reviewer commented that our manuscript “is worthy of publication” and requested us to address the following points:
- “The experiments were only conducted in experimental cells. Therefore, do such mutations actually occur in the natural world?” To answer this question, in-depth sequencing analysis focusing on this region of viral genome is essential to establish functional correlation, which is beyond the scope of this report. Of serendipity, reviewer 1 pointed out that the mutations in this area are particularly high including the processing site. This suggests that the regulated precursor catalysis reported in our paper might occur in the real world. We added one sentence to address this speculation (lines 303-6).
- “If virus production is hindered when such mutations occur, virus production will eventually cease. Therefore, it may be meaningless to link mutations in the adjacent amino acids in the Mpro precursor to the development of antiviral drugs.” We fully agree with this viewpoint, and it is not our intention to link mutations in the N-terminal processing site for antiviral development, either. Our analyses revealed an interesting aspect of precursor catalysis, which is subject to complex regulation by p1 mutations. We added more description to improve clarity (lines 303-6).
- “Does the development of antiviral drugs refer to drugs that cause mutations in the adjacent amino acids? Molnupiravir was reported to have an antiviral effect by changing the amino acid sequence, and there is a powerful paper showing that such an approach is effective.” As discussed in the previous point, it is not our intention to develop antivirals causing mutation at the processing site. We revised the manuscripts to improve clarity.
- “If Mpro with adjacent mutations amino acids is synthesized and incorporated into the virus, and if the resulting protein is functional, it will create a completely new mutant virus, which is very interesting from a clinical level.” A good point for molecular mechanistic characterization, which is beyond the scope of this work. We added this point for discussion as suggested.
“If there are similar examples for other viruses, it would be good to introduce them…If such a mechanism can produce viral resistance, clarification is awaited from a molecular perspective regarding the structural differences the adjacent amino acids produce in wild-type Mpro.” We indeed have unpublished data from HIV protease precursors showing association of some P1 mutations with drug resistance development. However, it is a little premature to reveal these unpublish results.